# Influence of COVID-19 on female sex workers in Dar es Salaam, Tanzania: A mixed-methods analysis

**Marianna Balampama[1], Damien de Walque[2], William H. Dow[3,4], Rebecca Hémono[3]***

1 Independent scholar, Dar-es-Salaam, Tanzania, 2 Development Research Group, The World Bank, Washington, DC, United States of America, 3 School of Public Health, University of California at Berkeley, Berkeley, California, United States of America, 4 National Bureau of Economic Research, Cambridge, Massachusetts, United States of America

* rebeccahemono@berkeley.edu

**Data Availability Statement:** Data cannot be shared publicly because of the sensitive nature of the activity (sex work) of the study participants as

## Abstract

This study investigates how the landscape of sex work in Dar es Salaam, Tanzania, evolved in the context of the COVID-19 epidemic. Using a mixed-methods approach, the analysis triangulates data from quantitative and qualitative sources to quantify shifts in income, demand, and client frequency and describe female sex workers' perspectives on their work environment. The COVID-19 restrictions introduced in early 2020 resulted in dramatic decreases in sex work income, leading to extreme financial vulnerability, food insecurity, and challenges in meeting other basic needs such as paying rent. However, in a 2021 follow-up survey, sex workers reported the summer of 2021 as a key turning point, with the demand for sex work rebounding to closer to pre-pandemic levels. Notably, despite the average number of unique weekly clients not yet having fully rebounded, by 2021 the price per client and the total monthly sex work income had returned to pre-pandemic levels. This may potentially be explained by an increased number of repeat clients, which represented a larger proportion of all clients during the COVID-19 pandemic.

## Introduction

COVID-19 has had profound effects on the global economy, health, and wellbeing of households worldwide [1,2]. Marginalized groups and individuals working in the informal sector, including female sex workers (FSW), have been disproportionately affected by the pandemic due to lower demand for their activity either because of fear of contagion or lockdowns, reduced financial protection, and a limited ability to navigate income shocks [3–5]. Previous assessments have found that in addition to disrupting sex work, COVID-19 lockdowns increased mental health challenges, food insecurity, and exacerbated barriers to accessing health care services including reducing HIV/STI prevention and treatment [6,7]. In addition, FSW were usually not covered by government social protection programs and were left with few economic prospects, increasing their vulnerability to violence [3,6,8].

The acute impacts of the COVID-19 pandemic, particularly the effects of "shelter-in-place" measures, curfews, and closures of businesses and other essential services in early 2020, have

well as given the sensitive and intimate nature of many of the variables collected (HIV and STI status, sexual behaviors, etc.). The corresponding author (or the World Bank Microdata Library https://microdata.worldbank.org/) can be contacted for requests of de-identified data.

**Funding:** We are grateful for financial support from the Strategic Impact Evaluation Fund (SIEF) and Knowledge for Change Program (KCP), managed by the World Bank(DdW). These funding sources were not involved in the design or conduct of the research. The funders had no role in study design, data collection and analysis, decision to publish, or preparation of the manuscript.

**Competing interests:** The authors have declared that no competing interests exist.

been well documented [7–13]. However, public health restrictions evolved throughout the different waves of the pandemic, and these changes in regulation likely play an important role in how FSW continue to be affected. Despite this, little is known about the economic impacts for FSW after the initial public health protection measures were eased or lifted, and how the landscape of sex work subsequently changed. We conducted a mixed-methods analysis of the effects of the COVID-19 pandemic on FSW in Dar es Salaam, Tanzania to determine how their activities shifted over the course of the pandemic and to understand how FSW were affected economically.

## Study setting: COVID-19 in Tanzania

The Director-General of the World Health Organization (WHO) declared COVID-19 a global pandemic on March 11, 2020 [14]. The first case of COVID-19 was reported in Tanzania five days later [15] and was followed by WHO-recommended public health restrictions [16], including government closures of schools and universities. Public gatherings were banned with the exception of churches and mosques, many businesses were temporarily or permanently closed, and travelers were required to quarantine for a 14-day period.

Government reporting of COVID-19 data ended in April 2020 after 509 cumulative cases and 21 deaths [17]. The late president, John Magufuli, declared in June 2020 that COVID-19 had been eradicated from Tanzania, and the government reduced testing and monitoring efforts and ordered schools to reopen [17,18]. Case counts, hospitalization, and mortality data were unavailable; however, informal reports suggest that there were multiple waves of respiratory illness and deaths in the subsequent months. Despite this, the government did not endorse mask-wearing or vaccination [18].

In March 2021, President Magufuli died and was replaced by President Samia Suluhu Hassan, who announced plans to develop a coronavirus task force and urged the citizens of Tanzania to take preventive measures, including social distancing [17]. The President warned of a third wave in June 2021, though masks and vaccination efforts were not publicly endorsed. The Ministry of Health, Community Development, Gender, Elderly and Children (MOHCD-GEC) then changed course, submitting a request to the global COVAX program, an initiative that is co-led between Gavi, the Vaccine Alliance, WHO, and the Coalition for Epidemic Preparedness Innovations to improve global equitable access to COVID-19 vaccines. Government officials subsequently reported 176 incident cases and 29 deaths during the "third wave", recommended mask use, precautionary measures, and began a vaccination campaign [19]. By June 1, 2022, 33,928 cumulative cases and 803 deaths had been reported,[15] likely a significant underestimation of the true toll.

## Methods

### Study population and design

The *Rewarding STI Prevention and Control in Tanzania II* ("RESPECT II") study [20] was conducted from May 2018 to January 2022 in Dar es Salaam, Tanzania, which has an estimated FSW population of 155,450 and an HIV prevalence of 26% for FSW [21]. The primary aim of the study was to investigate the effect of a lottery intervention on combined HIV/STI incidence among FSW. In May 2018, 2,206 participants were enrolled using respondent driven sampling (RDS) within each arm to identify FSW from various locations (bars, brothels, street). RDS is a chain referral sampling method designed to recruit hard-to-reach and hidden populations. Individuals meeting the following eligibility criteria were eligible to participate: 1) female; 2) exchanged sexual intercourse for money in the past 6 months; 3) HIV-negative at enrollment; 4) aged 18 years or older; 5) not currently pregnant; 6) lived in Dar es Salaam for past three

months and plan to remain living in Dar es Salaam for at least next 2 years; 7) have a cell phone able to receive text messages; 8) able to adequately grant informed consent; 9) in possession of a valid coupon (obtained through Respondent Driven Sampling).

Participants were randomized in a 1:1 ratio to receive standard HIV prevention services (testing, counseling, and weekly text messages on safe sex practices), or standard HIV prevention services plus entry in a random lottery rewarding safe sex as proxied by negative test for curable STIs.

## Data collection

Quantitative surveys were administered by trained female enumerators on tablets and paper questionnaires with all enrolled participants at two time points (wave 1, August 2018—February 2019 and wave 3, June 2021—January 2022). Surveys included questions on employment and income, client history, sexual behavior, HIV/STI risks, mental health, and gender-based violence. Surveys at the end of the study period included additional questions about changes in work and sexual activity related to COVID-19. An additional survey was administered via phone to a subset of participants in the control arm only (wave 2, July-October 2020), and included a reduced set of survey questions focused on wealth and income, income shocks related to COVID-19, food security, and sexual history.

Qualitative interviews were conducted by trained female interviewers with a random sample of 20 wave 3 participants using a semi-structured interview guide to elicit views on the intervention and to understand the effects of COVID-19 on employment, income, client type, and work conditions. The interview guide was informed by preliminary findings from the wave 3 quantitative surveys which indicated that participants had been experiencing changes in income and work activity related to COVID-19, thus the interviews served to gather additional perspectives and insights into the experiences of FSW during COVID-19. Interviews were approximately 60 minutes in duration and audio recorded. Data were collected until theme saturation was reached [22].

All data were collected in Kiswahili by trained research assistants at Innovations for Poverty Action Tanzania.

## Data analysis

Quantitative data were analyzed in STATA version 15.1. Descriptive statistics were generated to: 1) assess the demographic characteristics of participants at enrollment (wave 1, 2018) and to compare the overall study population with those who were reached for phone surveys (wave 2, 2020) and at the end of the study period (wave 3, 2021); 2) examine self-reported data on employment and income before and during COVID-19. The following variables were used to describe the sociodemographic characteristics of the study participants and assess their economic security and working landscape pre- and during COVID-19 using univariate analyses (means, standard deviation and frequencies): age, education, marital status, living arrangements, number of children, total monthly income, monthly income from sex work, location of sex work, number of clients, mean price for sex work with clients, and food insecurity. Linear first difference regression with robust standard errors and chi-square tests were used to generate p-values for differences in total income, sex work income, price per client, and clients per week, food security, and place of work pre- and during COVID-19. Balanced panels were generated to compare outcomes by the study population in each wave. Total and sex work income in waves 1 and 2 were adjusted for inflation (10.4% wave 1, 3.7% wave 2) [23]. Food security was expressed both as a binary variable and categorized by frequency.

Interviews were transcribed verbatim and translated from Kiswahili to English. The transcripts were analyzed in Dedoose version 8.3.17. Coding was done iteratively, with codes initially developed based on the interview guide and revised throughout the coding process as concepts emerged.

## Ethics

Approval for this study was obtained from the Institutional Review Boards at the University of California, Berkeley (2015-08-7849) and National Institute for Medical Research in Tanzania (NIMR/HQ/R.8a/Vol.IX/2770). Written informed consent was obtained from study participants.

## Results

At enrollment (wave 1), the mean age of participants was 26 years and more than half (58.9%) had completed at least primary education (Table 1). Most were never married/single (75.8%), had children (70.6%), and living alone or with family (77.9%).

Among the subset of control participants selected for phone surveys (wave 2), 346 (59%) were successfully reached. Half of all participants were lost to follow up (49.4%) at 24 months (wave 3). Participants reached for wave 3 were slightly older in age, and more had children and were previously partnered (divorced/separated) than the overall sample at enrollment.

In-depth interview participants were also slightly older in age than the full sample at enrollment. Most had completed at least primary education (95.0%), were not married (75.0%), and had children (90.0%). Half reported living with family (50.0%) (S1 Table).

**Table 1. Baseline demographic characteristics of study participants in the RESPECT II trial.**

|  | Wave 1 (2018) (n = 2,206) | Wave 2 (2020) (n = 346) | Wave 3 (2021) (n = 1,117) | p-value (Wave 1,3) | p-value (Wave 1,2) |
|---|---|---|---|---|---|
| **Age** | 26 (6.7) | 28 (6.6) | 28 (7.1) | <0.01 | <0.01 |
| **Education** |  |  |  | 0.57 | 0.15 |
| None or some primary | 226 (10.2%) | 28 (8.1%) | 122 (10.9%) |  |  |
| Primary complete | 1,300 (58.9%) | 203 (59.7%) | 662 (59.3%) |  |  |
| Some secondary | 282 (12.8%) | 40 (11.6%) | 141 (12.6%) |  |  |
| Secondary complete | 398 (18.0%) | 75 (21.7%) | 192 (17.2%) |  |  |
| **Marital status** |  |  |  | <0.01 | <0.01 |
| Never married/single | 1,672 (75.8%) | 243 (70.2%) | 805 (72.1%) |  |  |
| Divorced/separated | 455 (20.6%) | 82 (23.7%) | 271 (24.3%) |  |  |
| Married | 32 (1.5%) | 11 (3.2%) | 15 (1.3%) |  |  |
| Widowed | 30 (1.4%) | 9 (2.6%) | 14 (1.3%) |  |  |
| Cohabitating | 16 (0.7%) | 1 (0.3%) | 11 (1.0%) |  |  |
| **Living arrangement** |  |  |  | 0.17 | 0.95 |
| Alone | 928 (42.0%) | 147 (42.5%) | 472 (42.3%) |  |  |
| With partner | 59 (2.7%) | 10 (2.9%) | 36 (3.2%) |  |  |
| With family | 792 (35.9%) | 120 (34.7%) | 408 (36.5%) |  |  |
| With friends/other | 427 (19.4%) | 69 (19.9%) | 201 (18.0%) |  |  |
| **Has children** | 1,558 (70.6%) | 257 (74.3%) | 831 (74.4%) | <0.01 | 0.10 |

Missing data: Marital status (all respondents, n = 1).

Data are presented as mean (SD) and frequency (%).

p-values represent comparisons in baseline characteristics between Wave 1 and 3 participants and Wave 1 and Wave 2 participants.

## COVID-19 exposure and precautions

Interview participants were aware of and knowledgeable about COVID-19 transmission and viewed themselves at high risk, despite limited availability of information or national data on COVID-19. Many had concerns about being exposed given their close proximity to clients.

> *"I am concerned because the nature of our work involves close physical contact between two people, and doing so tends to increase the risk of transmission of this pandemic."*

When asked about exposure reduction strategies, most participants expressed that their work was not conducive to wearing masks or taking other precautions against COVID-19.

> *"There are some precautions that don't align with the nature of my work. For instance, wearing face masks, or washing hands regularly during sex is not possible and this is why I feel at risk."*

One participant noted that clients viewed sex workers without masks as less desirable and preferred to engage with those who wore masks. Therefore, some women opted to wear masks when initially interacting with clients in an effort to appear less risky and secure more work.

> *"They believed that women who weren't wearing face masks were not only at high risk but also insensible. So we had to wear face masks not only to protect ourselves but also to attract clients."*

## Shifts in work landscape

**Place of work.**   Interviewees shared that they were required to seek work in new locations during the initial phase of the pandemic. With bars closed and gatherings prohibited in March 2020, many resorted to connecting and arranging to meet with clients via phone.

> *"We were often chased away from our working places because the government had prohibited gatherings and the nature of our work had us standing in groups beside the road to wait for clients. So it became difficult for us to continue working in this environment. Not only that, but also sitting in bars was prohibited too. We therefore had to work through the phone, as in waiting for the client to contact us through the phone which made work become difficult."*

> *"At first, we used to go to bars to find clients, but during the pandemic it was really challenging because gatherings were prohibited. There were few people at the bars. Some bars were closed. So, if you met a client you exchanged numbers."*

Survey data reveal significant changes in the place of work from 2018 to 2021 and suggest that the ways of meeting clients may have shifted despite many venues having reopened. Seeking work in bars decreased from 40.9% in 2018 to 34.7% in 2021 and in brothels from 33.1% to 28.0% (Table 2). Participants were more likely to meet clients on the street (6.7% to 12.4%) and in "other locations" (19.3% to 24.9%).

**Type of clients.**   When bars and brothels closed, meeting new clients became more difficult and engaging with repeat clients became more common. Some explained how they would exchange phone numbers with clients to reach them more easily.

> *"During the pandemic, most of the times I saw repeat clients, unlike new clients, as they were the ones who I could at least persuade to meet me again. I used to ask for their contacts so that*

**Table 2. Economic security and work landscape pre- and during COVID-19 with balanced panels.**

| | Balanced panel in waves 1 and 3 | | | Balanced panel in waves 1 and 2 | | | Balanced panel in waves 2 and 3 | | |
|---|---|---|---|---|---|---|---|---|---|
| | Wave 1 2018 (n = 1,117) | Wave 3 2021 (n = 1,117) | p-value | Wave 1 2018 (n = 346) | Wave 2 2020 (n = 346) | p-value | Wave 2 2020 (n = 214) | Wave 3 2021 (n = 214) | p-value |
| Total income (USD/mo) | 123.1 (145.7) | 117.6 (93.2) | 0.28 | 120.2 (98.9) | 88.1 (98.0) | <0.01 | 92.5 (113.4) | 125.4 (109.1) | <0.01 |
| Sex work income* (USD/mo) | 121.6 (146.4) | 119.0 (78.6) | 0.66 | 113.9 (91.7) | 79.5 (55.4) | <0.01 | 84.2 (60.9) | 127.1 (83.7) | <0.01 |
| Clients/week | 8.6 (7.0) | 6.2 (5.5) | <0.01 | 8.4 (6.7) | 4.8 (3.2) | <0.01 | 5.0 (3.4) | 6.3 (5.9) | <0.01 |
| Food insecurity | 14.8 | 26.1 | <0.01 | 13.6 | 45.4 | <0.01 | 43.0 | 24.3 | <0.01 |
| Place of work** | | | | N/A | | | | | |
| Street | 6.7 | 12.4 | <0.01 | | | | | | |
| Bar | 40.9 | 34.7 | | | | | | | |
| Brothel | 33.1 | 28.0 | | | | | | | |
| Other | 19.3 | 24.9 | | | | | | | |
| Mean price/client | 6.6 (9.6) | 6.2 (5.5) | 0.27 | | | | | | |

Missing data: Total income (Wave 1 & 3 balanced panel, n = 21, Wave 1 & 2 balanced panel, n = 11) sex work income (Wave 1 & 3 balanced panel n = 296, Wave 1 & 2 balanced panel, n = 97, Wave 2 & 3 balanced panel n = 70), place of work (Wave 1 & 3 balanced panel n = 156), mean price per client (Wave 1 & 3 balanced panel n = 124), clients per week (Wave 1 & 3 balanced panel1 n = 126, Wave 1 & 2 balanced panel n = 155, Wave 2 & 3 balanced panel, n = 23)).

Data are presented as mean (SD) and frequency (%).

*Among only participants who report sex work as main source of income.

**Place of work and mean price per client not measured at in Wave 2.

*when I needed them, it was easier for me to reach them. It was easier persuading repeat clients than new clients."*

This trend continued throughout 2021. Although there were no restrictions on gatherings and businesses were open, participants complained that the bars were no longer crowded and that it was difficult to meet clients there as they had done previously.

*"Clients have declined, the bars are not crowded enough. I only have a few clients who have my phone number who can call me if they need my service."*

Some interviewees suggested that repeat clients were more common because they felt more comfortable seeking their services due to more trust and less fear of exposure.

*"The clients that used to contact me more often were those who are so close to me and they were the ones who I was willing to meet during the pandemic."*

*"I saw repeat clients more. New clients were threatened by the pandemic. In fact, the repeat clients were comfortable coming to me because they knew they could pay whatever they had, and I would understand them because I was used to them."*

## Changes in the demand and price of sex work

When the government implemented restrictions in March 2020, job losses and high unemployment rates led to low demand for sex work. Participants described how their clients struggled to meet their basic needs, and thus could not afford to pay for sex.

*"Most people stopped going to work, and same people are the ones that we depend on to earn our living. The fact that they stopped working made us not see them more often*

*because the little they earned wasn't enough to afford meeting their basic, family, and sexual needs."*

In turn, many interviewees explained how the low demand for work left them unable to meet their own needs, such as paying for food or housing. Women who relied primarily on sex work for their income were heavily affected, and some reported food insecurity.

*"My life became so hard that it reached a time when I couldn't afford my meals. There were times that I was compelled to have meals twice a day because to earn 10,000 TZS per day was not easy."*

*"The pandemic has brought intense effects especially to us who depended on sex work and had no other source of income. The pandemic therefore had us suffer economically because we couldn't get clients making it difficult for me to meet my basic needs such as food."*

Survey data demonstrate that food insecurity was highest while restrictions were in place, significantly increasing from 14.5% in 2018 to 43.0% in 2020 and subsequently decreasing to 24.3% in 2021 after restrictions had been lifted (Fig 1).

Participants also reported that many clients who sought sex work attempted to negotiate on the price, arguing that they could not afford the prices that they were previously paying prior to COVID-19. Some participants shared that they were initially reluctant to negotiate however ultimately opted to accept a lower price rather than forgo the work.

*"Clients bargained on the price that we gave them because they complained on being broke. So instead of paying 10,000 [TZS] for instance, they'd pay 5,000 [TZS]. Therefore, the price decreased compared to how it was before the pandemic."*

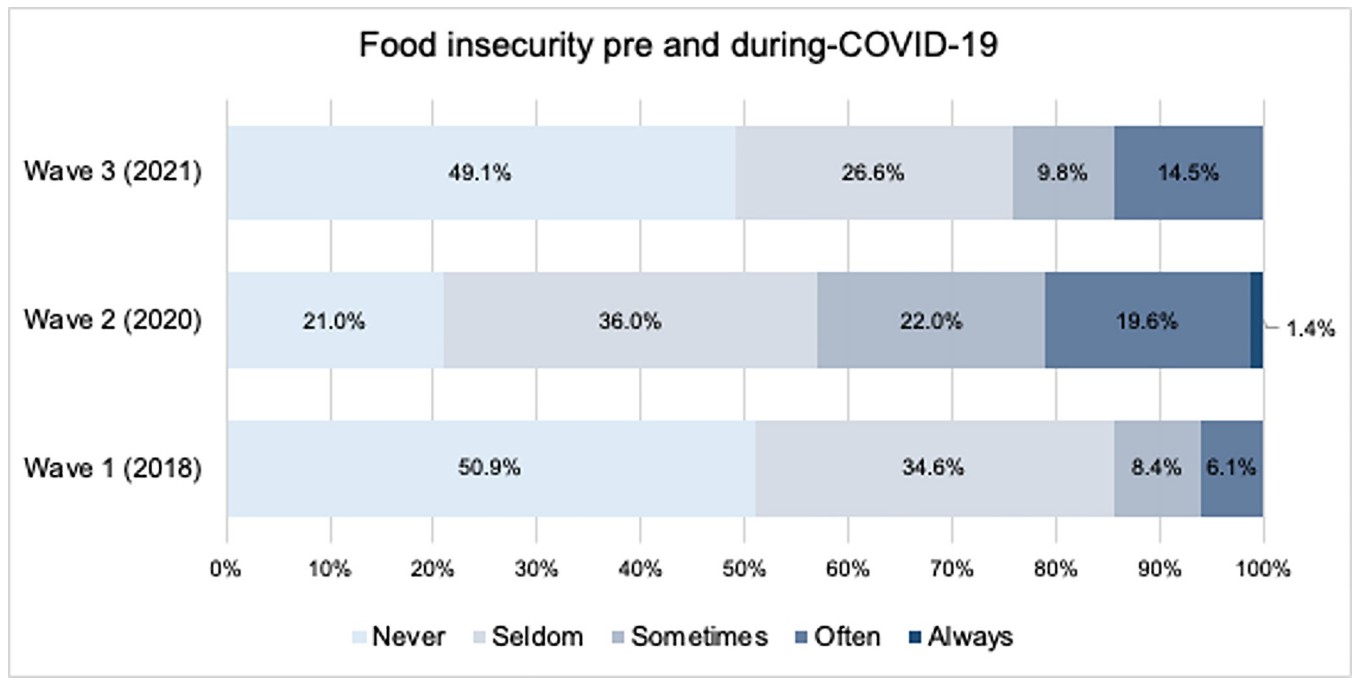

**Fig 1. Food security among balanced panel of 2018, 2020, and 2021 participants (n = 214).**

*"I had to change how I charged my clients. This was because most of them complained on being broke because of the pandemic, so I always had to bargain on my charges. I had to agree with them because if I did not, they left for other women and my biggest fear by then was to go back home from work without having earned anything. Before the pandemic, I used to charge them 30,000 TZS but during the pandemic it dropped up to between 10,000 to 5000 TZS."*

While negotiating was still common for some, most participants explained how prices had begun to return to pre-pandemic levels. Survey data indicate that there was a decrease in the average price per client from 6.6 USD in 2018 to 6.2 USD in 2021, though the difference was non-significant.

Participants also reported a significantly lower average number of clients per week in 2021 (6.2) than in 2018 (8.6). The lowest levels were experienced in 2020, with participants reporting an average of 4.8 per week. However, most interviewees explained that they started to observe higher demand for work when restrictions on gatherings and workplace closures were lifted.

*"In the first wave of the pandemic, there were less clients but then as time went on, the demand started getting back to normal, but they have not yet reached the normal level."*

*"The biggest changes happened during the time when the government issued a declaration of closure of schools, banning gatherings. . .it has at least started getting back to normal but not yet the same as it was before the pandemic. In late June to July 2021, I started noticing that the work was at least improving."*

Many described June and July 2021 as a key turning point when they began to notice their work returning to normal, which coincides with the shift in the government's approach to addressing COVID-19. Some participants now viewed the pandemic as being "over" and felt that their income was no longer affected.

*"It started improving in July to August, and by September, October 2021 it had reached to normal up to date.*

*"Since June 2021, last year. I started noticing that the pandemic was over because I earned a lot of money."*

Similar results emerged from the survey data, which indicate that sex work income has recovered from the initial price shocks that occurred at the beginning of 2020. Among participants who reported sex work as their primary source of income, the total monthly income from sex work reported in surveys significantly increased from 84 USD to 127 USD per month from 2020 to 2021. Total income also recovered by 2021 and was similar to income levels prior to the pandemic.

## Discussion

This study found that by late 2021 FSW in Dar es Salaam, Tanzania had mostly recovered from the economic insecurity following the initial phase of the COVID-19 pandemic in 2020. The restrictions introduced in early 2020 resulted in dramatic decreases in sex work income, leading to extreme financial vulnerability, food insecurity, and challenges in meeting other basic needs such as paying rent. However, participants reported that the demand for sex work substantially increased in summer 2021. Notably, despite decreases in the average number of unique weekly clients, the price per client and the total monthly sex work income had returned to pre-pandemic levels. This may potentially be explained by an increased number of repeat

clients, which started to represent a larger proportion of all clients during COVID-19, even after restrictions were lifted. Additionally, this study revealed that the pandemic changed the landscape of sex work in many ways, including the places to meet new clients and ways of meeting and engaging with clients. In 2021, participants were still not meeting clients as regularly in bars and brothels and were more likely to meet clients on the street.

Our results are consistent with findings from studies in rural and urban Kenya which demonstrated sharp decreases in transactional sex activity, income, and numbers of clients in the early phases of the pandemic [24,25]. Other research conducted in India, Ukraine, Kenya, and Zimbabwe also found that many FSW struggled to meet basic needs such as pay rent and experienced increased food insecurity during this period [3,26]. In addition, our findings are supported by previous studies which have highlighted how patterns of work have shifted over the pandemic, with fewer FSW going to bars and nightclubs, and more working in street based settings or independently in apartments.[3,13]

This study advances our understanding of how the landscape of sex work evolved during the first two years of the COVID-19 pandemic. Using a mixed-methods approach, we triangulate data from quantitative and qualitative sources to quantify shifts in income, demand, and client frequency, and describe FSW perspectives on their current work environment. While previous studies have documented the acute effects of COVID-19 restrictions on the livelihoods and wellbeing of FSW, there has been limited understanding of how these effects subsequently evolved. Our results underscore that there were severe economic impacts that ensued after restrictions were put into effect at the onset of the COVID-19 pandemic, and reinforce the need for social protection mechanisms for key vulnerable populations and informal workers [27,28]. However, our study also demonstrates how demand, income, and economic security improved after restrictions were lifted and pandemic concerns receded, underscoring that informal employment in the context of COVID-19 is not a static experience and continues to fluctuate as restrictions shift and personal risk tolerance changes. In the Tanzanian context, the government's limited public health response and approach to reopening may also have mitigated income shocks for FSW and reduced COVID-19 related fear and stigma [29]; nonetheless, the health implications in terms of COVID morbidity and mortality of this approach should not be underestimated.

There are some important limitations to this study. While efforts were made to support participants in reflecting on their work during surveys and interviews, poor recall may have affected the ability of participants to correctly report their experiences. In addition, quantitative and qualitative data collection occurred at different time points. Wave 2 surveys were conducted among a subset of participants with access to a mobile phone and do not include perspectives from the full sample, limiting our ability to explore longitudinal effects over critical time points. There was also substantial attrition in wave 3 due to study delays and increased migration related to COVID-19. In addition, new government regulations were introduced which required registration of all mobile phone numbers using biometrics and national ID cards. These regulations resulted in many individuals losing access to their mobile phone numbers, which was our primary mode of contact with study participants. Therefore, it is possible that the balanced sample may be less representative of the overall study sample.

In conclusion, this study improves our understanding of how the experiences of FSW evolved over the course of an extended pandemic shock. We demonstrate that while ways of meeting clients and working changed, the financial hardship experienced early in the pandemic largely dissipated in the second year. However, despite improvements in economic security and wellbeing, there is a critical need to integrate social protection mechanisms for FSW and other key vulnerable populations as part of public health emergency preparedness

and response measures to ensure that sex workers are not further marginalized and are able to meet basic needs during crisis periods.

## Supporting information

**S1 Table. Demographic characteristics of in-depth interview participants.**
(DOCX)

## Acknowledgments

We are grateful to Kathleen Beegle, Emiko Masaki and Gil Shapira for useful comments. We would like to thank Pooja Suri for excellent research assistance, the study team at Innovations for Poverty Action Tanzania for their support with data collection and all study participants for their time participating in our research. The findings, interpretations, and conclusions expressed in this paper are entirely those of the authors. They do not necessarily represent the views of the authors' institutions or funders or of the International Bank for Reconstruction and Development/World Bank and its affiliated organizations, or those of the Executive Directors of the World Bank or the governments they represent. All errors are our own.

## Author Contributions

**Conceptualization:** Marianna Balampama, Damien de Walque, William H. Dow, Rebecca Hémono.

**Data curation:** Rebecca Hémono.

**Funding acquisition:** Damien de Walque, William H. Dow.

**Investigation:** Marianna Balampama, Damien de Walque, William H. Dow.

**Methodology:** Marianna Balampama, Damien de Walque, William H. Dow, Rebecca Hémono.

**Project administration:** Marianna Balampama, Damien de Walque, William H. Dow.

**Resources:** Damien de Walque, William H. Dow.

**Supervision:** Marianna Balampama, Damien de Walque, William H. Dow.

**Validation:** Marianna Balampama, Damien de Walque, William H. Dow.

**Visualization:** Rebecca Hémono.

**Writing – original draft:** Rebecca Hémono.

**Writing – review & editing:** Marianna Balampama, Damien de Walque, William H. Dow.

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
