## [Decision Letter · Decision Letter 0]

17 Mar 2024

PONE-D-23-41981Influence of COVID-19 on Female Sex Workers in Dar es Salaam, Tanzania: A Mixed-Methods AnalysisPLOS ONE

Dear Dr. de Walque,

Thank you for submitting your manuscript to PLOS ONE. After careful consideration, we feel that it has merit but does not fully meet PLOS ONE’s publication criteria as it currently stands. Therefore, we invite you to submit a revised version of the manuscript that addresses the points raised during the review process.

Please submit your revised manuscript by May 01 2024 11:59PM**.** If you will need more time than this to complete your revisions, please reply to this message or contact the journal office at plosone@plos.org. Please include the following items when submitting your revised manuscript:A rebuttal letter that responds to each point raised by the academic editor and reviewer(s). You should upload this letter as a separate file labeled 'Response to Reviewers'.A marked-up copy of your manuscript that highlights changes made to the original version. You should upload this as a separate file labeled 'Revised Manuscript with Track Changes'.An unmarked version of your revised paper without tracked changes. You should upload this as a separate file labeled 'Manuscript'.

We look forward to receiving your revised manuscript.

Kind regards,

Joseph Gregory Rosen

Academic Editor

PLOS ONE

 [We are grateful for financial support from the Strategic Impact Evaluation Fund (SIEF) and Knowledge for Change Program (KCP), managed by the World Bank(DdW). These funding sources were not involved in the design or conduct of the research.].  

Additional Editor Comments:

I commend the authors for this well-written mixed methods paper that addresses shifting dynamics in sex work over the course of the COVID-19 pandemic in urban Tanzania. While I would have preferred to identify two independent peer reviewers to assess the quality of the manuscript, I recognize the review process has already been extensively delayed through fruitless efforts to secure additional reviewers. Thus, I have rendered a decision on the manuscript based on the assessment of one reviewer and myself. In addition to the comments provided by the first reviewer, I advise the authors to pay particular attention to the following two concerns: (1) Please define how all quantitative measures were captured and assessed in the study methods; (2) What were drivers of attrition/lost to follow-up reported in wave 3, and were these attributed to the COVID-19 pandemic? I am confident the authors can address these comments expeditiously and improve the quality of the submission. Thank you for the opportunity to review this paper.

Reviewers' comments:

Reviewer's Responses to Questions

**Comments to the Author**

1. Is the manuscript technically sound, and do the data support the conclusions?

Reviewer #1: Yes

2. Has the statistical analysis been performed appropriately and rigorously? 

Reviewer #1: Yes

3. Have the authors made all data underlying the findings in their manuscript fully available?

Reviewer #1: Yes

4. Is the manuscript presented in an intelligible fashion and written in standard English?

Reviewer #1: Yes

5. Review Comments to the Author

Reviewer #1: In this manuscript, the authors used a mixed-methods approach to examine changes in female sex worker’s work environment in the context of changing policies in Tanzania during the COVID-19 pandemic. This paper is well-written and succeeds in capturing shifting dynamics in the sex work landscape during the pandemic. I have just a couple of minor points to offer for the authors to consider.

- Can the authors provide some information on who conducted the interviews and administered the surveys (e.g., gender, credentials, experience/training)?

- How were interview participants sampled (e.g., convenience sampling, other)? Can the authors describe the characteristics of the interview participants?

- Can the authors comment on the extent to which they reached data saturation?

6. PLOS authors have the option to publish the peer review history of their article (what does this mean?). If published, this will include your full peer review and any attached files.

Reviewer #1: No

---

## [Author Response · Author response to Decision Letter 0]

1 May 2024

Many thanks for the very useful comments and suggestions.

Please find below our point-by-point responses (in bold) to each comment.

Best regards,

The authors.

Changes made. 

 [We are grateful for financial support from the Strategic Impact Evaluation Fund (SIEF) and Knowledge for Change Program (KCP), managed by the World Bank(DdW). These funding sources were not involved in the design or conduct of the research.]. 

This role of the funder been added to the manuscript (p. 16) and in the cover letter.

We have updated our explanation of the restrictions to sharing the data as follows:

Initial explanation: Data cannot be shared publicly because of the sensitive nature of the activity (sex work) of the study participants. Data are available from the authors for researchers who meet the criteria for access to confidential data.

Revised explanation: Data cannot be shared publicly because of the sensitive nature of the activity (sex work) of the study participants as well as given the sensitive and intimate nature of many of the variables collected (HIV and STI status, sexual behaviors, etc.). The corresponding author (or the World Bank Microdata Library https://microdata.worldbank.org/) can be contacted for requests of de-identified data.

The ethics statement appears in the Methods section (pg. 5).

Additional Editor Comments:

I commend the authors for this well-written mixed methods paper that addresses shifting dynamics in sex work over the course of the COVID-19 pandemic in urban Tanzania. While I would have preferred to identify two independent peer reviewers to assess the quality of the manuscript, I recognize the review process has already been extensively delayed through fruitless efforts to secure additional reviewers. Thus, I have rendered a decision on the manuscript based on the assessment of one reviewer and myself. In addition to the comments provided by the first reviewer, 

I advise the authors to pay particular attention to the following two concerns: (1) Please define how all quantitative measures were captured and assessed in the study methods; 

Response: Details on how all the quantitative measures were captured and assessed have been added to Methods, Data analysis (p. 5):

“The following variables were used to describe the sociodemographic characteristics of the study participants and assess their economic security and working landscape pre- and during COVID-19 using univariate analyses (means, standard deviation and frequencies): age, education, marital status, living arrangements, number of children, total monthly income, monthly income from sex work, location of sex work, number of clients, mean price for sex work with clients, and food insecurity.” 

(2) What were drivers of attrition/lost to follow-up reported in wave 3, and were these attributed to the COVID-19 pandemic? 

Response: We have added a more detailed explanation of the possible reasons we experienced wave 3 attrition in the Discussion (p. 14-15)

“There was also substantial attrition in wave 3 due to study delays and increased migration related to COVID-19. In addition, new government regulations were introduced which required registration of all mobile phone numbers using biometrics and national ID cards. These regulations resulted in many individuals losing access to their mobile phone numbers, which was our primary mode of contact with study participants. Therefore, it is possible that the balanced sample may be less representative of the overall study sample:”

I am confident the authors can address these comments expeditiously and improve the quality of the submission. Thank you for the opportunity to review this paper.

Reviewers' comments:

Reviewer's Responses to Questions

Comments to the Author

1. Is the manuscript technically sound, and do the data support the conclusions?

Reviewer #1: Yes

2. Has the statistical analysis been performed appropriately and rigorously? 

Reviewer #1: Yes

3. Have the authors made all data underlying the findings in their manuscript fully available?

Reviewer #1: Yes

4. Is the manuscript presented in an intelligible fashion and written in standard English?

Reviewer #1: Yes

5. Review Comments to the Author

Reviewer #1: In this manuscript, the authors used a mixed-methods approach to examine changes in female sex worker’s work environment in the context of changing policies in Tanzania during the COVID-19 pandemic. This paper is well-written and succeeds in capturing shifting dynamics in the sex work landscape during the pandemic. I have just a couple of minor points to offer for the authors to consider.

- Can the authors provide some information on who conducted the interviews and administered the surveys (e.g., gender, credentials, experience/training)?

Response: Personnel for the field work was entirely composed of females, as all counseling and interviewing should be done with personnel of the same gender as participants. Interviewers and field workers were trained for each specific study procedures: quantitative survey, qualitative interviews and specimen collection. HIV and STI testing, treatment, and all counselling was performed by a partner organization which had qualified and trained staff with several years of experiences in providing HIV/AIDS services to the population in Dar-es-Salam. 

We have added under the data collection subsection, under methods, p. 4 the fact that all enumerators/interviewers were trained and female. Since this manuscript does not include data from HIV and STI testing, we do not include further details on the qualification of the staff conducting this part of the data collection. 

- How were interview participants sampled (e.g., convenience sampling, other)? Can the authors describe the characteristics of the interview participants?

Response: . The female sex workers were recruited using respondent-driven sampling (RDS) within each arm, a chain referral sampling method designed to recruit hard-to-reach and hidden populations. We have further described this method in the study population and design subsection under Methods p. 3. The eligibility criteria for the participants are detailed in the same section and the socio-demographic of the participants are reported in Table 1 and described under results (pp 5-6).

- Can the authors comment on the extent to which they reached data saturation?

Response: Interviews were conducted until data saturation was reached (i.e., no new information would have been obtained by collecting more data (Morse 1995)). This information has been added under Data collection, p. 4.

6. PLOS authors have the option to publish the peer review history of their article (what does this mean?). If published, this will include your full peer review and any attached files.

Do you want your identity to be public for this peer review? For information about this choice, including consent withdrawal, please see our Privacy Policy.

Reviewer #1: No

---

## [Editor Report · Decision Letter 1]

6 May 2024

Influence of COVID-19 on Female Sex Workers in Dar es Salaam, Tanzania: A Mixed-Methods Analysis

PONE-D-23-41981R1

Dear Dr. de Walque,

We’re pleased to inform you that your manuscript has been judged scientifically suitable for publication and will be formally accepted for publication once it meets all outstanding technical requirements.

Kind regards,

Joseph Gregory Rosen

Academic Editor

PLOS ONE

Additional Editor Comments (optional):

Thank you to the authors for carefully attending to all reviewer and editorial comments. I commend the authors on on an impactful publication.

---

## [Editor Report · Acceptance letter]

21 May 2024

PONE-D-23-41981R1 

PLOS ONE

Dear Dr. de Walque, 

I'm pleased to inform you that your manuscript has been deemed suitable for publication in PLOS ONE. Congratulations! Your manuscript is now being handed over to our production team.

Kind regards, 

on behalf of

Dr. Joseph Gregory Rosen 

Academic Editor

PLOS ONE